# Influence of Surface Modification of MK-40 Membrane with Polyaniline on Scale Formation under Electrodialysis

**DOI:** 10.3390/membranes10070145

**Published:** 2020-07-07

**Authors:** Marina A. Andreeva, Natalia V. Loza, Natalia D. Pis’menskaya, Lasaad Dammak, Christian Larchet

**Affiliations:** 1Physical Chemistry Department, Faculty of Chemistry and High Technologies, Kuban State University, 149 Stavropolskaya st., 350040 Krasnodar, Russia; nata_loza@mail.ru (N.V.L.); n_pismen@mail.ru (N.D.P.); 2Institut de Chimie et des Matériaux Paris-Est (ICMPE) UMR 7182 CNRS, Université Paris-Est, 2 Rue Henri Dunant, 94320 Thiais, France; dammak@u-pec.fr (L.D.); larchet@u-pec.fr (C.L.)

**Keywords:** ion-exchange membrane, polyaniline, surface modification, mineral scaling, voltammetry, chronopotentiometry

## Abstract

A comprehensive study of the polyaniline influence on mineral scaling on the surface of the heterogeneous MK-40 sulfocationite membrane under electrodialysis has been conducted. Current-voltage curves and chronopotentiograms have been obtained and analyzed for the pristine MK-40 membrane and the MK-40 membrane which is surface-modified by polyaniline. The study of the electrochemical behavior of membranes has been accompanied by the simultaneous control of the pH of the solution outcoming from the desalination compartment. The mixture of Na_2_CO_3_, KCl, CaCl_2_, and MgCl_2_ is used as a model salt solution. Two limiting states are observed on the current-voltage curve of the surface-modified membrane. There is the first pseudo-limiting state in the range of small values of the potential drop. The second limiting current is comparable with that of the limiting current for the pristine membrane. It is shown that chronopotentiometry cannot be used as a self-sufficient method for membrane scaling identification on the surface-modified membrane at high currents. A mineral scale on the surfaces of the studied membranes has been found by scanning electron microscopy. The amount of precipitate is higher in the case of the surface-modified membrane compared with the pristine one.

## 1. Introduction

To date, methods of electromembrane technology are effective, environmentally cleanest, and cost-effective [1,2]. Electrodialysis (ED) processes are widely implemented for brine concentrations in sea-salt production [2], the food industry [3,4], the extraction of precious or toxic substances, such as organic acids [5,6], wastewater treatment, especially for the removal of heavy metals [7,8], and production of acids and bases [9]. ED is based on the selective migration of ions through ion-exchange membranes (IEMs) under the action of applied electric field as the driving force. Low-energy consumption and high-current efficiency are the advantages of the ED process. However, despite the efforts of many other researchers to optimize ED performance for various applications, there are several weaknesses that restrict ED usefulness, involving selectivity, membrane fouling, and mineral scaling.

Mineral scaling on IEMs happens when salts from the solution precipitate and settle on the membrane surface and inside the ion-conductive pathways (pores) of the membrane [10]. Scaling is closely related to the development of one of the coupled effects of the concentration polarization in the electromembrane system, namely, water splitting [11,12]. pH variations of the solution close to the interface boundary result in the creation of conditions where precipitation of sparingly soluble salts occurs. The precipitate occurs when the solute concentrations exceed the solubility of the sparingly soluble solids [13]. The multivalent ions such as barium, calcium, magnesium, ferric, bicarbonate, and sulfate are the major scaling ions. The presence of mineral scales on the membrane surface and inside the membrane pores reduces the membrane working area and operation lifetime, causes additional resistance to the solution flux, and mass transfer.

Another coupled effect of concentration polarization, namely, electroconvection, hinders the process of water splitting [14,15,16]. Thus, it is possible to reduce the negative effect of scaling by enhancing electroconvection [17,18,19]. It was shown that the coating of a homogeneous Nafion^®^ film on the surface of a heterogeneous MK-40 membrane leads to the electroconvection intensification and water splitting decrease due to a relatively more hydrophobic and less electrically inhomogeneous surface compared to the original MK-40 membrane, and as a result, to scaling mitigation [19].

The most used methods for reducing the membrane scaling are changing regimes of electrodialysis treatment, mechanical action, pretreatment using pressure-driven membrane processes, cleaning agents, and the modification of IEMs [10]. Despite the fact that researches show the successful tendency for IEM clogging mitigation by means of electrodialysis reversal and pulsed electrical field [20], IEMs scaling is still a limiting factor for the wide construction of ED.

Surface topology of the IEMs also influences the scale formation. The rougher the membrane surface, the more susceptible it is to scaling due to the presence of more active sites for surface nucleation [13,18,19]. The smoother membrane surface results in weaker scale-membrane adhesion. Thereby the development of membranes with improved scaling resistance is a promising alternative for scaling mitigation.

Another way of membrane surface modification leading to the monovalent membrane selectivity is used for membrane clogging mitigation [21]. It is known from the literature that the formation of a thin oppositely charged layer on the surface of IEMs could improve the membrane selectivity towards the transfer of monovalent ions [22]. The monovalent ion selectivity stems from the kinetic effect of electrostatic repulsion of the charged surface layer towards divalent cations in the solution [22]. Thus, the transfer of multicharged ions from the desalination compartment to the concentration compartment reduces, mitigates, or prevents scaling in the concentration compartment. Some studies have also found that the above modification helps drastically decrease the peptide fouling during electrodialysis as a result of changes in the nature of the electrostatic interaction between proteins and the membrane surface [21]. It has been shown that polymerization of polyaniline (PANI) on the surface of the cation-exchange membrane results in the appearance of selectivity for the transfer of singly charged ions [23,24] and the decrease of water transport number [25]. At the same time, as a result of modifying the membrane with PANI, there was no significant decrease in the permselectivity of the cation exchange membrane [26]. Nevertheless, the formation of an internal bipolar interface between the cation-and anion-exchange layers led to catalytic water splitting that could stimulate the rate of scale formation on the IEM [27]. Potentially, these membranes may be used as monovalent-cation-selective ones. However, their behavior has not been studied from the point of view of possible scaling. In this context, the main goal of the article is to study how the PANI modification of heterogeneous membranes affects the scaling process.

## 2. Materials and Methods

### 2.1. Membranes

Heterogeneous ion-exchange MK-40 and MK-40/PANI membranes are tested in the study. The heterogeneous cation-exchange MK-40 membrane, manufactured by UCC “Shchekinoazot” (urban locality Pervomaysky, Tula oblast, Russia), is a mixture of the polyethylene and cation exchange resin KU-2-8 based on sulfonated polystyrene crosslinked by divinylbenzene. To ensure the mechanical strength, the membrane is reinforced with a nylon mesh. MK-40 membranes are used in the electrodialysis concentration and demineralization of electrolyte solutions [28,29,30,31]. The closest analog of this membrane is the Ralex-CMH membrane produced by MEGA a.s. (Straz pod Ralskem, Czech Republic). The properties of the MK-40 membrane are described elsewhere, for example in [32,33,34,35,36]. The MK-40/PANI membrane is obtained by synthesizing PANI on the surface of the MK-40 membrane under conditions of electrodiffusion of the monomer and the oxidizer in an external electric field [37]. Thus, the MK-40/PANI composite has an asymmetric structure—the MK-40 membrane is on the one side, and the PANI layer is on the other side. The surfaces of the tested membranes are illustrated in Figure 1.

The heterogeneous anion-exchange commercial MA-41 membrane, manufactured by Shchekinoazot (Russia) and the MK-40 membrane are used as the auxiliary membranes in the chronopotentiometry and voltammetry measurements. The functional groups of the MA-41 membrane are quaternary ammonium groups.

### 2.2. Analysis Methods

#### 2.2.1. Membrane Thickness

The membrane thickness, *L*, is measured with a Micrometer MK 0-25 (Model 102, Chelyabinsk, Russia). The membrane thickness values are averaged from ten measurements at different locations on the effective surface of each membrane.

#### 2.2.2. Membrane Electrical Conductivity

The membrane conductivity is determined by the difference method. The membrane electrical conductivity, *κ*, could be calculated as
(1)$$\kappa =\frac{L}{R_{m}S},$$
where *S* and *L* are the membrane area and thickness, respectively, *R_m_*—the electric resistance of the membrane.

The electrical resistance is measured with a specially designed cell coupled to the Immitance meter (RLC) E7-21 (MNIPI, Minsk, Republic of Belarus), and 0.1 M NaCl reference solution is used. According to the procedure developed by Lteif et al. [38], the electric resistance of the membrane is calculated according to:(2)$$R_{m}=R_{m+s}-R_{s},$$
where *R_m+s_* and *R_s_* are the electrical resistance of the solution with membrane and without membrane, respectively.

#### 2.2.3. Contact Angle Measurements

Contact angles, *θ*, of the wet membranes are measured by the sessile drop method in the sodium form [39]. Data are treated using the ImageJ software.

#### 2.2.4. Scanning Electron Microscopy

Visualization of the membrane surface covered by the precipitate is made by a scanning electron microscope (Merlin, Carl Zeiss Microscopy GmbH, Oberkochen, Germany) equipped with an energy dispersive spectrometer. The energy dispersive spectrometer conditions are 6 kV accelerating voltage with a 9.9-mm working distance. The dried membrane samples are coated with a thin layer of platinum in order to make them electrically conductive and to improve the quality of the microscopy photographs.

### 2.3. Electrodialysis Cell and Experimental Setup

The investigations are carried out in the same flow-through four-compartment electrodialysis cell, which is used for the membrane modification [19]. The cell comprises the tested MK-40 or MK-40/PANI membrane and two auxiliary membranes. The intermembrane distance in the cell compartments is 0.64 cm; the membrane area exposed to the current flow is 4 cm^2^. The anode and the cathode are platinum-polarizing electrodes. The experimental setup involves two closed loops containing a model salt solution and a 0.04 M NaCl electrolyte solution. The model solution circulates across the central desalination compartment—the 0.04 M NaCl solution—through an auxiliary and two electrode compartments in parallel, where the solution flow rate through each compartment is 30 mL min^−1^. The model salt solution is composed of Na_2_CO_3_ (1000 mg L^−1^), KCl (800 mg L^−1^), CaCl_2_*2H_2_O (4116 mg L^−1^), and MgCl_2_*6H_2_O (2440 mg L^−1^). In this solution, the Mg^2+^/Ca^2+^ molar concentrations ratio is 2/5, and the total concentration of MgCl_2_ and CaCl_2_ is 0.04 M to ensure membrane scaling formation [40]. The pH of the solution is adjusted to 6.5 by adding HCl. The mineral composition of this solution corresponds approximately to that of trice the concentrated milk. Each closed loop is connected to a separated external plastic reservoir, allowing continuous recirculation.

### 2.4. Protocol

The procedures for measuring I-V curves and chronopotentiograms (ChPs) of MK-40 and MK-40/PANI membranes are described in [39]. A KEITHLEY 220 current source is used to supply the current between the polarizing electrodes. The potential drop (PD) across the membrane under study, ∆*φ*, is measured using Ag/AgCl electrodes. These electrodes are placed in the Luggin capillaries. The Luggin tips are installed at both sides of the membrane under study in its geometric center at a distance of about 0.5 mm from the surface. PD is registered by a HEWLETT PACKARD 34401A multimeter. The I-V curves are recorded when the current is swept from 0 to 10.5 mA cm^−2^ at a scan rate of 0.8 μA s^−1^; the PD remains < 3 V. The chronopotentiometric measurements are made in the range of current densities from 0.2 *i*_lim_ to 1.5 *i*_lim_, where *i*_lim_ is the experimentally determined limiting current density. The experiment is carried out at 20 °C. All I-V curves and ChPs for the MK-40/PANI membrane are measured in such a way that the PANI layer of the composite is oriented toward the desalination compartment.

## 3. Results and Discussion

### 3.1. Physico-Chemical Characteristics of Ion-Exchange Membranes

As shown in Figure 1, the green PANI chains are observed only on the particles of ion-exchange resin at the membrane surface, excluding polyethylene. The PANI layer within the MK-40/PANI membrane has weak anion-exchange properties. Thus, its formation leads to a bipolar interface, in this case, the main layer is the non-modified part of the pristine cation-exchange membrane. The thickness of MK-40/PANI membrane rises by 0.02 mm compared to the the pristine MK-40 membrane (Table 1). The conductivity of MK-40/PANI is about 16% lower than that of the MK-40 membrane. These data correlate with earlier results [41]. Shkirskaya et al. had studied the influence of the PANI layer both on the homogeneous and on the heterogeneous surface of sulfocationite membranes on their electric conductivity in an NaCl solution. A decrease in the electric conductivity in 10% of the MK-40/PANI composite was observed in the diluted solutions (less than 0.3 M). A further increase in the solution concentration leveled out these differences. However, for homogeneous MF-4SK membranes (Russian analog of Nafion^®^), the electric conductivity decreased significantly after modification by PANI. The authors suggested that membrane heterogeneity is a more significant factor than the polyaniline synthesis conditions. Along with this, the contact angle of MK-40 and MK-40/PANI is nearly the same, close to 55°, indicating the invariability of the hydrophilic properties of the membrane surface. The contact angle does not change because polyethylene covers about 80% of both MK-40 and MK-40/PANI membranes.

### 3.2. Voltammetry

There are three regions on the I-V curve of monopolar IEMs, which are generally distinguished in the literature [12,16]. An initial linear region is followed by a plateau of limiting current density, and then by a region of higher growth of current density. When approaching *i*_lim_, the electrolyte interfacial concentration is nearly zero that initiates coupled effects of concentration polarization—current-induced convection (electroconvection and gravitational convection) and water splitting [12]. The shape of the I-V curve is influenced by many factors. The plateau of limiting current density becomes less appreciable, the slope appears, and the length of the plateau decreases when the concentration of the solution and the stirring rate increase [42]. Similar effects are observed in solutions of complex compositions, for example, containing surfactants [43] or multicharged ions [44]. In this case, the method of numerical differentiation is used to evaluate the limiting current, and the value of the limiting current is determined as an extremum on the curves expressed in d(∆*φ*)/d*i*-*i*_av_ coordinates.

Figure 2 shows the I-V curves obtained for MK-40 and MK-40/PANI membranes in the model salt solution and the corresponding dependence of the pH of the desalinated solution. In case of the 

MK-40 membrane, the shape of the I-V curve is the typical one. Water splitting at the depleted membrane interface, begins after reaching *i*_lim_. The I-V curve obtained for the MK-40/PANI membrane has two inflection points (Figure 2).

The first inflection point is in the region of small PD about 300 mV. Apparently, it is related to the depletion of the bipolar interface between the cation- and anion-exchange layers within the cation exchange particles on the membrane surface. Under the action of external electric field, the cations migrate from the bipolar interface into the bulk of the cation-exchange layer; the anions migrate from this interface into the bulk of the anion-exchange layer. At a certain current density, the ion concentration at this interface becomes sufficiently low leading to an essential increase in the membrane resistance. The latter is seen by an appearing plateau on the I-V curve in the range of current densities close to 4 mA cm^−2^. This critical current density, which is caused by the depletion of the bipolar interface, may be called the pseudo-limiting current density, *i*_pseudo-lim_. When the concentration of the salt ions at the depleted bipolar interface becomes sufficiently low, water splitting occurs similar to that which takes place in the bipolar membranes [11,45]. This is confirmed by a change in the pH of the solution coming out of the desalination compartment at *i* ≈ 4 mA cm^−2^ (Figure 2, green dashed line). This process produces H^+^ and OH^-^ ions, which are additional current carriers and whose emergence result in decreasing the membrane resistance. This is confirmed by the fact that the derived I-V curve of the modified membrane goes down after reaching *i*_pseudo-lim_, which indicates a decrease in the resistance of the electromembrane system (Figure 3). Thus, the bipolar interface between the cation and anion exchange layers causes a significant increase in the water splitting rate compared to the MK-40 membrane (Figure 2). Similar phenomena were observed for the asymmetric bipolar membranes, in which the thickness of one of the layers is much larger [46].

As the current continues to increase, the ion concentration at the external membrane surface decreases to a low value that causes a new increase in the system resistance in the case of the MK-40/PANI membrane. Since the anion exchange layer has a very small thickness compared to the cation exchange layer, this leads to the fact that through this layer, cations are transported by the diffusion mechanism. In this case, we see the development of concentration polarization according to the classical type for monopolar membranes. This state relates to the “classical” limiting current density, *i*_lim_, observed in case of conventional monopolar IEMs [12]. The presence of two limiting current densities can be seen more clearly on the derived I-V curves for the studied membranes in the model solution shown in Figure 3.

The values of both limiting current densities may be determined by the point of intersection of the tangents drawn to the linear parts of the I-V curve on the left and on the right of the region, where a sharp change in the rate of the current growth with potential drop occurs (Figure 2). These values for both limiting current densities are present in Table 2.

At the same time, there are no two extrema on the derived I-V curves for asymmetric bipolar membranes compared to the modified membrane studied in this work. This may be because PANI synthesis has been carried out directly on the surface of the IEM, and the localization of the modifier is the particles of cation-exchange resin. As a result, a structure is formed where the anion and cation layers are much closer to each other compared to the classic and asymmetric bipolar membranes. Classic bipolar membranes are mechanically composed of anion and cation exchange layers. A thin anion exchange layer is deposited on the cation substrate in the asymmetric bipolar membranes. However, a more significant change in the pH of the desalination solution in the case of the MK-40/PANI membrane compared to the MK-40 membrane indicates continued generation of H^+^ and OH^-^ ions at the internal bipolar interface. In examining the polarization behavior of the MK-40/PANI membrane, a change in the shape of the I-V curve is noted during the experiments. The amount of electricity flowing through the system required to change the shape of the I-V curve is equal to 2000 C. There is no *i*_pseudo-lim_ on the derived I-V curve and the value of *i*_lim_ becomes higher in the case of the used MK-40/PANI membrane compared to the fresh MK-40/PANI membrane (Figure 4). Probably, the clearly expressed bipolar interface of the composite is vanishing; therefore, the pseudo-limiting current disappears on the I-V curve. The resistance of the ohmic region decreases, and the plateau region becomes longer and more pronounced.

### 3.3. Chronopotentiometry

The ChP of the pristine MK-40 membrane and its derivative measured in the 0.04 M MgCl_2_ solution at an overlimiting current density (1.9 *i*_lim_) is presented in Figure 5. The curve behavior has a classical form for monopolar IEMs at *i* ≥ *i*_lim_, discussed in the literature [47,48]. When the current starts to flow, there is a speedy growth of PD related to the ohmic potential drop, ∆*φ*_ohm_, over the membrane and two adjacent solutions, and there is no influence of concentration polarization. The value of PD does not increase significantly during one experimental run at a current density less the limiting value. However, the shape of ChP changes differently at *i* ≥ *i*_lim_. The electrolyte concentration at the depleted membrane/solution interface decreases by degrees with time, the process is governed by electrodiffusion. As a result, there is a drastic rise in PD on the ChP. The electrolyte concentration at the depleted membrane/solution interface gets low enough by a certain time called the transition time, *τ*, [47,48]. This time corresponds to the appearance of an additional mechanism of ion transport, namely, the current-induced convection. Therefore, the growth rate of PD falls, and PD tends to a quasi-steady state value, ∆*φ*_st_.

Figure 6 shows the ChPs for the MK-40 and MK-40/PANI membranes measured in the model salt solution at different current densities and the corresponding time-dependence of the pH of the desalinated solution. To enhance the presentation of the obtained ChPs, ∆*φ*_ohm_ is deducted from the measured value of Δ*φ* at a corresponding time and ChPs are represented at Δ*φ*’ vs. *t* coordinates. Δ*φ*’ is the reduced potential drop and can be calculated as follows:(3)$$\Delta \varphi ’\left(t\right)=\;\Delta \varphi \left(t\right)-\Delta \varphi_{\mathrm{ohm}}.$$

In case of the MK-40 membrane, the chronopotentiogram has a classical form. There is no transition time in the current density below *i*_lim_, and the pH of the desalinated solution does not change within the error (Figure 6a). In case of the MK-40/PANI membrane, the transition time is observed on the ChP at a current density higher that *i*_pseudo-lim_ but lower than *i*_lim_ (Figure 6a). Also, there is a noticeable alkalization of the solution in the desalination compartment in the case of the MK-40/PANI membrane (Figure 6a). This point corresponds to the appearance of an additional mechanism of ion transport—the emergence of H^+^ and OH^-^ ions due to water splitting at the internal bipolar interface occurring when the ion concentration at this boundary becomes sufficiently low. This is consistent with the results of voltammetry.

Also, the transition time is observed on the ChP for MK-40/PANI membrane at *i* = *i*_lim_, but there is a more significant change in the pH of the desalinated solution compared to the lower current densities (Figure 6b). It is also noted that PD slowly increases and does not reach a quasi-steady state value in time for the MK-40/PANI membrane (Figure 6b). A similar shape of the ChP was observed during the sedimentation of mineral salts on the surface and inside the pores of IEMs [49]. It could be assumed that the similar behavior of the ChPs in our study is associated with sedimentation on the membrane surface. A small amount of scale on the surface of MK-40/PANI membrane has been found after the experiment by visual inspection.

The PD slowly increases and does not reach a quasi-steady state value in time at currents higher than the limiting current for the MK-40 membrane (Figure 6c), which indicates the scale formation. However, the scale has not been detected in the measuring cell during the experimental run and on the membrane surface after the experiment by visual inspection. It should be noted that the pH of the desalinated solution does not change within the error in this case. The slow growth of PD in time on the ChP for the MK-40/PANI membrane is observed only at *i* = *i*_lim_, when the solution in the desalination compartment is significantly alkalized (Figure 6b).

At further growth in current density, the PD on the ChP for the MK-40/PANI membrane increases during the first 100–200 sec, and then gradually decreases reaching a quasi-steady state (Figure 6c,d). Also, the rate of change in the pH of the desalinated solution increases compared to that observed at *i* = *i*_lim_. H^+^ and OH^−^ ions appear in the system due to water splitting at the internal MK-40/PANI boundary. There is an apparent contradiction between the behavior of the membrane presented in Figure 6b,c. Even though the shape of the ChP in Figure 6c does not indicate the appearance of mineral scale, the muddy solution in the desalination compartment and precipitate on the surface-modified membrane was visually observed at *i* > *i*_lim_. Water splitting at the internal MK-40/PANI boundary at high currents has a dominant effect on the shape of the ChP compared to the mineral scaling on the membrane. The value of the PD decreases due to the appearance of new charge carriers (H^+^ and OH^−^ ions) instead of its increase due to scale formation.

The surfaces of the studied membranes have been checked by scanning electron microscopy after chronopotentiometry to validate the present assumptions. The analysis of the obtained results of scanning electron microscopy confirms the precipitate on both the pristine and surface-modified membranes (Figure 7). The precipitate, having a lamellar structure typical of calcite, as well as more complex polycrystalline forms of calcium carbonate, were found [50,51]. Moreover, the mineral scale covers the part of the working surface area of the MK-40 membrane (Figure 7a–c) and its entire surface of the MK-40/PANI membrane (Figure 7d–f). 

According to energy-dispersive analysis, the precipitate on the membrane surfaces is a mixture of calcium carbonate and magnesium hydroxide with a predominance of calcium carbonate (Figure 8). There were similar findings obtained by C. Casademont et al. [52,53,54,55].

## 4. Conclusions

The scale formation on the surface of two ion-exchange membranes, a heterogeneous MK-40 and its modification MK-40/PANI, during electrodialysis of a solution containing the scale-forming cations, Ca^2+^ and Mg^2+^, was investigated by the method of voltammetry and chronopotentiometry.

The I-V curve had two inflection points in the case of the MK-40/PANI membrane in comparison with the MK-40 membrane. The first pseudo-limiting current density is in the region of small displacements of the potential from equilibrium. It was related to the depletion of the bipolar interface within the membrane. As the current continued to increase, the ion concentration at the external membrane surface decreased to a low value that caused a new increase in the system resistance, and the “classical” limiting current density was observed. The disappearance of the pseudo-limiting current with the simultaneous increase in the value of the limiting current density was observed for the MK-40/PANI membrane at the long-term usage.

The precipitate was detected at a current density of more than 0.8 *i*_lim_ for the MK-40/PANI, and more than *i*_lim_ for the MK-40 membrane. However, at current densities less than 0.8 *i*_lim_, no scaling was observed for the MK-40/PANI membrane. This was evidenced by the growth of the potential drop across the membrane at *i* = *i*_lim_ for MK-40/PANI and at *i* = 1.2 *i*_lim_ for the MK-40 membranes instead of reaching the quasi-steady state. However, the shape of the chronopotentiogram for the MK-40/PANI membrane changed at currents above the limit. At the same time, the water splitting rate at the MK-40/PANI bipolar boundary increased. All these facts indicated that the water splitting reaction was a determining factor in the electrochemical behavior of the membrane system than mineral scaling at currents above the limit. Despite the fact that the amount of precipitate is higher on the surface of the MK-40/PANI membrane, the values of the limiting current and electrical conductivity of the MK-40/PANI membrane were comparable with the values of the MK-40 membrane. Thus, it is possible to recommend the MK-40/PANI composite membrane for monovalent-ion-selective electrodialysis at the underlimiting current densities.

## Figures and Tables

**Figure 1 membranes-10-00145-f001:**
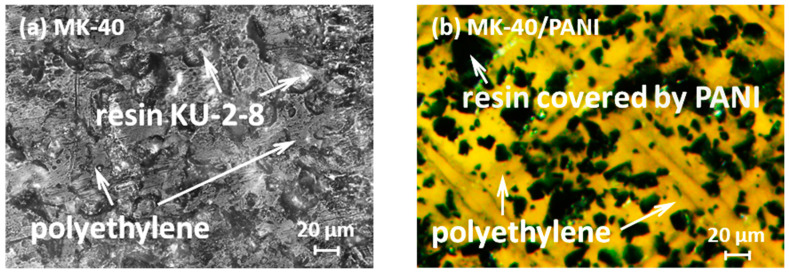
Image of the surface of swollen (**a**) MK-40 and (**b**) MK-40/PANI membranes obtained with a Zeiss AxioCam MRc5 light microscope.

**Figure 2 membranes-10-00145-f002:**
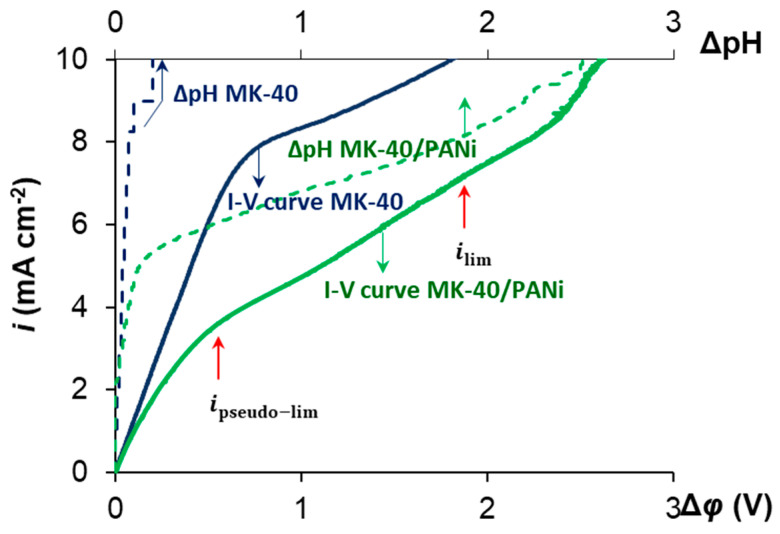
I-V curves for MK-40 and MK-40/PANI membranes in the model solution and corresponding dependence of the pH of desalinated solution.

**Figure 3 membranes-10-00145-f003:**
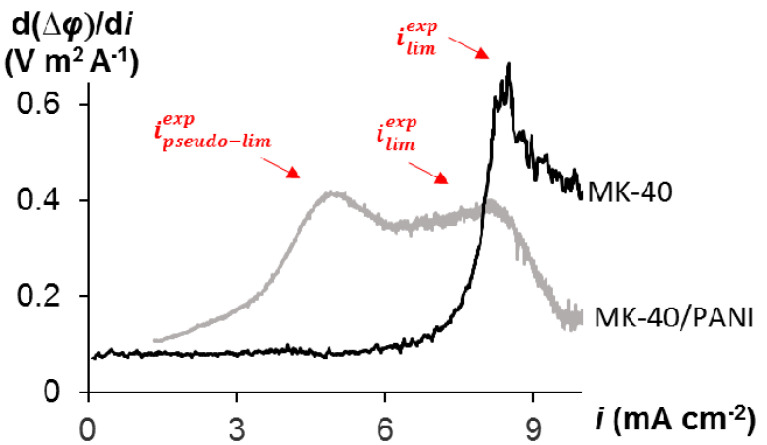
Derived I-V curves, d(∆*φ*)/d*i*, for MK-40 and MK-40/PANI membranes in the model solution at the corresponding current density.

**Figure 4 membranes-10-00145-f004:**
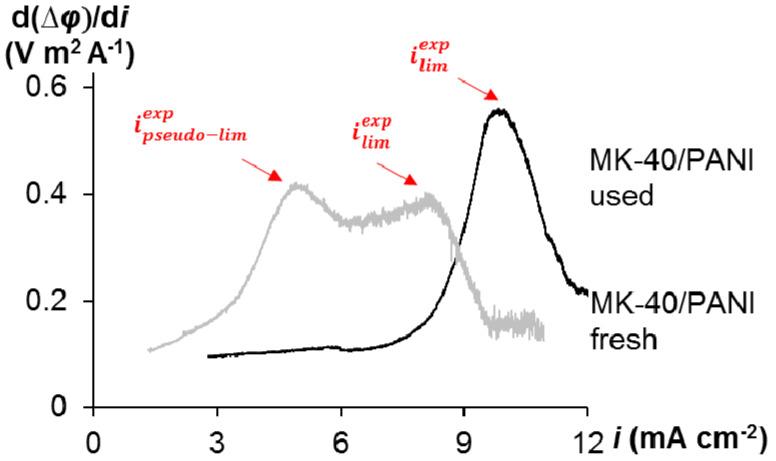
Derived I-V curves, d(∆*φ*)/d*i*, for fresh and used MK-40/PANI membranes.

**Figure 5 membranes-10-00145-f005:**
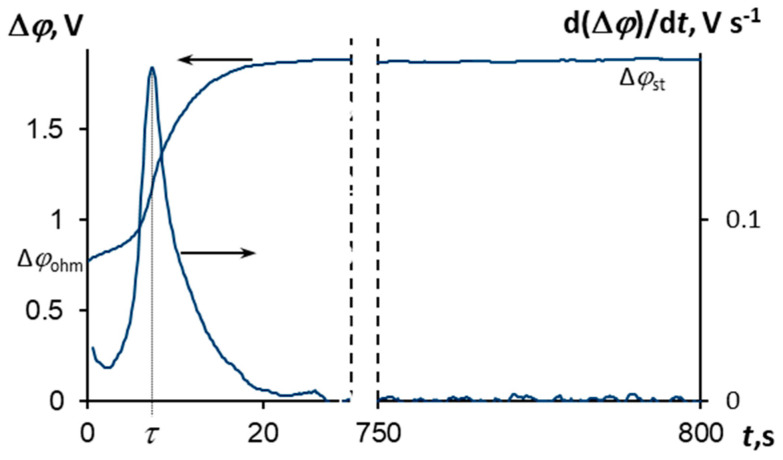
Chronopotentiogram of the MK-40 membrane in the 0.04 M MgCl_2_ solution, ∆*φ* vs. *t*, and its derivative, d(∆*φ*)/d*t* vs. *t*, in the model solution at *i* = 1.9 *i*_lim_. ∆*φ*_ohm_ is the ohmic potential drop just after the current is switched on; ∆*φ*_st_ is the potential drop at the quasi-steady state; *τ* is the transition time.

**Figure 6 membranes-10-00145-f006:**
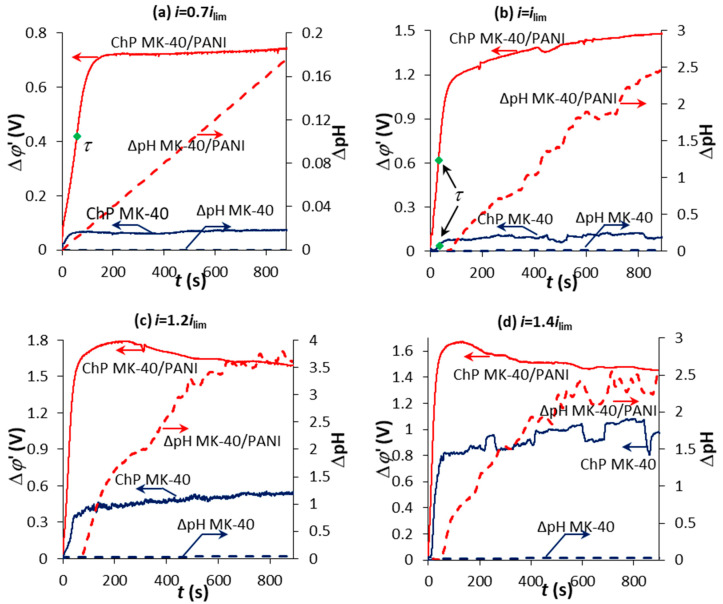
Chronopotentiograms of MK-40 and MK-40/PANI membranes in the model solution and the corresponding time-dependence of the pH of the desalinated solution at *i* = 0.7 *i*_lim_ (**a**); *i* = *i*_lim_ (**b**); *i* = 1.2 *i*_lim_ (**c**); *i* = 1.4 *i*_lim_ (**d**).

**Figure 7 membranes-10-00145-f007:**
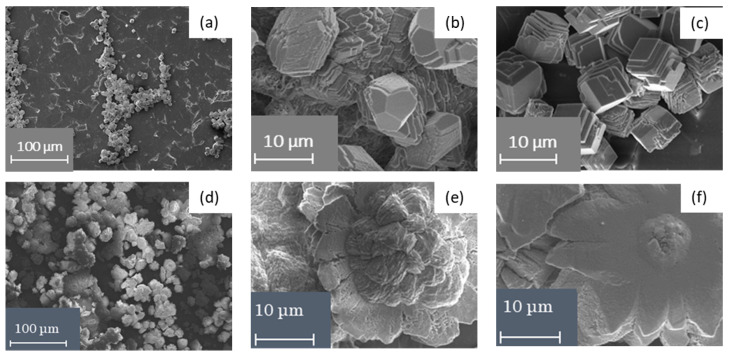
SEM images of the MK-40 (**a**–**c**) and MK-40/PANI (**d**–**f**) membrane surfaces facing the desalination compartment after 5 h of electrodialysis of model solution at *i* = 1.4 *i*_lim_.

**Figure 8 membranes-10-00145-f008:**
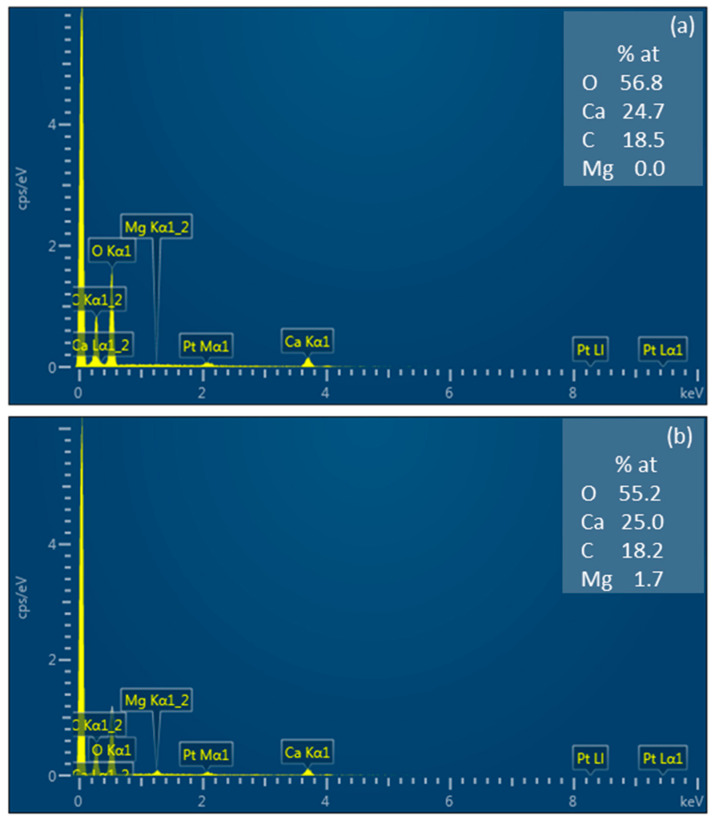
Energy dispersive X-ray spectroscopy of precipitate on the MK-40 membrane surface.

**Table 1 membranes-10-00145-t001:** The main properties of the MK-40 and MK-40/PANI membranes.

Membrane	*L*, mm	*κ*, S m^−1^	*θ*, °
MK-40	0.470 ± 0.005	0.36 ± 0.03	55 ± 3
MK-40/PANI	0.490 ± 0.005	0.30 ± 0.02	54 ± 3

**Table 2 membranes-10-00145-t002:** The values of limiting current densities for MK-40 and MK-40/PANI membranes in the model solution at 20 °C.

Membrane	*i*_pseudo-lim_, mA cm^−2^	*i*_lim_, mA cm^−2^
MK-40	-	7.7
MK-40/PANI	3.2	7.0

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
