# Peer review of "Influence of Surface Modification of MK-40 Membrane with Polyaniline on Scale Formation under Electrodialysis"

_membranes, 2020, doi:10.3390/membranes10070145_

Round 1
Reviewer 1 Report
This is an interesting study addressing the important topic of scaling in electrodialysis, which is a key problem. The authors propose a modification by polyaniline, which is original. It then also extends to scaling identification and the study of scale formation through identification methods. Therefore, the work has more than a single focus point.
Some suggetsions:
- Title - please consider if this really covers all the aspects of this study.
- What is the link between surface modification and selectivity? The author touch this topic but it would be worthwhile to probe more on this, also related to the literature that they refer to.
- The MK-40 membrane is rather less known. Is there more information, perhaps also referring to other studies where this membrane was used?
- How were the different parts of the membrane in Figure 1 identified?
- On page 4, 'This data correlate with the earlier results [29]': could the authors elaborate a little more?
- A bit disappointing that the work stops with characterization - what about performance/selectivity?
Author Response
Dear Reviewers,
We are grateful to Reviewers for their time and efforts aimed at improving our manuscript. We find your comments and recommendations quite useful. We have revised our manuscript according to your recommendations. The corrections are highlighted in yellow. Also, we have made some the corrections concerning the lexis. These corrections are highlighted in green. We have rewritten the first paragraph in Section 3.3 part to make it different from other published papers. The Reviewers’ comments and our responses are presented below.
With my best regards,
Marina Andreeva
Comments from the editors and reviewers:
-Reviewer 1
This is an interesting study addressing the important topic of scaling in electrodialysis, which is a key problem. The authors propose a modification by polyaniline, which is original. It then also extends to scaling identification and the study of scale formation through identification methods. Therefore, the work has more than a single focus point.
Some suggetsions:
- Title - please consider if this really covers all the aspects of this study.
Response
Thank you for this comment. We add the conditions for conducting scientific experiment in the title (line 2-4):
“Influence of surface modification of MK-40 membrane with polyaniline on scale formation under electrodialysis”.
- What is the link between surface modification and selectivity? The author touches this topic but it would be worthwhile to probe more on this, also related to the literature that they refer to.
Response
Thank you for this comment. It is known from the literature that the formation of a thin oppositely charged layer on the surface of ion-exchange membrane can improve the membrane selectivity towards the transfer of monovalent ions. The monovalent ion selectivity stems from the kinetic effect of electrostatic repulsion of the charged surface layer towards divalent cations in the solution. It has been shown that polymerization of polyaniline on the surface of the cation-exchange membrane results in the appearance of selectivity for the transfer of singly charged ions [Nagarale, R.K.; Gohil, G.S.; Shahi, V.K.; Trivedi, G.S.; Rangarajan, R. Preparation and electrochemical characterization of cation- and anion-exchange/polyaniline composite membranes. J. Colloid Interface Sci. 2004, doi:10.1016/j.jcis.2004.04.027; Sata, T. Composite Membranes Prepared from Cation Exchange Membranes and Polyaniline and Their Transport Properties in Electrodialysis. J. Electrochem. Soc. 1999, 146, 585, doi:10.1149/1.1391648]. At the same time as a result of modifying the membrane with polyaniline, there is no significant decrease in the permselectivity of the cation exchange membrane [Demina, O.A.; Shkirskaya, S.A.; Kononenko, N.A.; Nazyrova, E. V. Assessing the selectivity of composite ion-exchange membranes within the framework of the extended three-wire model of conduction. Russ. J. Electrochem. 2016, doi:10.1134/S1023193516040030]. So, we revised the fragment under consideration (line 65-83):
“Another way of membrane surface modification leading to the membrane monovalent selectivity is used for membrane clogging mitigation [21]. It is known from literature that the formation of a thin oppositely charged layer on the surface of IEM can improve the membrane selectivity towards the transfer of monovalent ions [22]. The monovalent ion selectivity stems from the kinetic effect of electrostatic repulsion of the charged surface layer towards divalent cations in the solution [22]. Thus, the transfer of multicharged ions from the desalination compartment to the concentration one reduces, mitigating or preventing scaling in the concentration compartment. Some studies have also found that the above modification helps drastically decrease the peptide fouling during electrodialysis as a result of changes in the nature of the electrostatic interaction between proteins and the membrane surface [21]. It has been shown that polymerization of polyaniline (PANI) on the surface of the cation-exchange membrane results in the appearance of selectivity for the transfer of singly charged ions [23,24] and the decrease of water transport number [25]. At the same time as a result of modifying the membrane with PANI, there is no significant decrease in the permselectivity of the cation exchange membrane [26]. Nevertheless, the formation of an internal bipolar interface between the cation- and anion- exchange layers lead to catalytic water splitting that can stimulate the rate of scale formation on the IEM [27]. Potentially, these membranes may be used as monovalent-cation-selective ones. However, their behavior has not been studied from the point of view of possible scaling. In this context, the main goal of the article is to study how PANI modification of heterogeneous membranes affects the scaling process.”
- The MK-40 membrane is rather less known. Is there more information, perhaps also referring to other studies where this membrane was used?
Response
Thank you for this comment. In this connection we add the information in the main text (line 90-93):
“MK-40 membranes are used in electrodialysis concentration and demineralization of electrolyte solutions [28–31]. The closest analogue of this membrane is Ralex-CMH membrane produced by MEGA a.s. (Czech Republic). The properties of MK-40 membrane is described elsewhere, for example in [32–36].”
- How were the different parts of the membrane in Figure 1 identified?
Response
Volodina et al. explored the element analysis of different parts on the MK-40 membrane by means of an electronic scanning microscope [Volodina, E.; Pismenskaya, N.; Nikonenko, V.; Larchet, C.; Pourcelly, G. Ion transfer across ion-exchange membranes with homogeneous and heterogeneous surfaces. J. Colloid Interface Sci. 2005, doi:10.1016/j.jcis.2004.11.017]. The existence of conducting (ion-exchange particles “protruding” from the wall) and nonconducting (polyethylene regions) phases was confirmed by electronic microscopy with element analysis. It was established that the film coating the larger part of the membrane consists of carbon atoms. The atoms constituting the ion exchange groups were not found. In contrast, the ion-exchange resin particles situated in the gaps of polyethylene film contained the atoms belonging to ionogenic groups of ion-exchange membrane equilibrated with NaCl solution.
- On page 4, 'This data correlate with the earlier results [29]': could the authors elaborate a little more?
Response
Thank you for this comment. We add some results in line 165-175:
“Shkirskaya et al. had studied the influence of PANI layer both on the homogeneous and on the heterogeneous surface of sulfocationite membranes on their electric conductivity in a NaCl solution. A decrease in the electric conductivity in 10% of the MK-40/PANI composite was observed in diluted solutions (less than 0.3 M). Further increase in the solution concentration leveled out these differences. However, for homogeneous MF-4SK membrane (Russian analog of Nafion®) the electric conductivity decreased significantly after modification by PANI. Authors suggested that the membrane heterogeneity is a more significant factor than the polyaniline synthesis conditions. Along with this the contact angle of MK-40 and MK-40/PANI is nearly the same, close to 55°, indicating the invariability of the hydrophilic properties of the membrane surface. The contact angle does not change because polyethylene covers about 80% of both MK-40 and MK-40/PANI membranes.”
- A bit disappointing that the work stops with characterization - what about performance/selectivity?
Response
We are in full agreement that the question about performance/selectivity is quite interesting. But selectivity is not investigated in this paper. We would like to stress that the aim of the article is to show how polyaniline modification of heterogeneous membranes affects the scaling process. We present all the results available now. More detail studies will be undertaken in future.

Reviewer 2 Report
This study compared the scaling formation between a pristine MK-40 membrane and modified MK-40/PANI membrane by the analysis of current- voltage curves and chronopotentiogram. The electrochemical characterizations were systematic performed and provided some informative references for characterization of other ion exchange membranes. A very interesting result was found that the modified MK-40/PANI membranes presented two limiting current density, one pseudo-limiting current density and other one classical limiting current density. This manuscript is generally well-documented. But there are several ambiguities should be clarified prior to potential publication.
Major concerns,
The purpose for a modified membrane is the performances of that modified membrane should be superior to the pristine membrane. But in this study, it seemed that the modified MK-40/PANI membrane exhibited less scaling resistance compared to the pristine membrane. The MK-40/PANI has a pseudo-limiting current density at i ≈ 4 mA cm-2, water dissociation would be initiated when such pseudo-limiting current density reached. It is much easier for the formation of scaling on MK-40/PANI compared to pristine membrane. In this case, what are the advantages of the modified MK-40/PANI membrane for scaling mitigation?
Minor concerns,
- In the abstract, the authors said “A comprehensive study of the polyaniline influence on mineral scaling”, but the influence of polyaniline on scaling was not even investigated. The authors just compared two kinds of membranes, how the effect of the addition of polyaniline on membrane scaling was not investigated.
- In line 29-30, methods of electromembrane technology are the most effective, environmentally cleanest and least costly. The sentence is overstated. Electromembrane is one of effective but should not be the most effective. Please modify such kind of statement.
- The composition of feed solution directly the membrane scaling. Why this kind of composition is chosen? According to the solubility product of CaCO3, it seemed that the feed solution is supersaturated. Please clarity it.
- For the current density, the authors sometimes used an exactly value such as i=5.0mAcm-2, but sometimes used a relative value of limiting current density i=1.4 ilim. Please make consistence of the description of the current density.
- In Line 298-300, “According to the energy dispersive analysis the precipitate on the membrane surfaces is a mixture of calcium carbonate and magnesium with a predominance of calcium carbonate”. How to make sure the precipitate is a mixture of calcium carbonate and magnesium with a predominance of calcium carbonate? Please provide evidences.
Author Response
Dear Reviewers,
We are grateful to Reviewers for their time and efforts aimed at improving our manuscript. We find your comments and recommendations quite useful. We have revised our manuscript according to your recommendations. The corrections are highlighted in yellow. Also, we have made some the corrections concerning the lexis. These corrections are highlighted in green. We have rewritten the first paragraph in Section 3.3 part to make it different from other published papers. The Reviewers’ comments and our responses are presented below.
With my best regards,
Marina Andreeva
Comments from the editors and reviewers:
-Reviewer 2
This study compared the scaling formation between a pristine MK-40 membrane and modified MK-40/PANI membrane by the analysis of current- voltage curves and chronopotentiogram. The electrochemical characterizations were systematic performed and provided some informative references for characterization of other ion exchange membranes. A very interesting result was found that the modified MK-40/PANI membranes presented two limiting current density, one pseudo-limiting current density and other one classical limiting current density. This manuscript is generally well-documented. But there are several ambiguities should be clarified prior to potential publication.
Major concerns,
The purpose for a modified membrane is the performances of that modified membrane should be superior to the pristine membrane. But in this study, it seemed that the modified MK-40/PANI membrane exhibited less scaling resistance compared to the pristine membrane. The MK-40/PANI has a pseudo-limiting current density at i ≈ 4 mA cm-2, water dissociation would be initiated when such pseudo-limiting current density reached. It is much easier for the formation of scaling on MK-40/PANI compared to pristine membrane. In this case, what are the advantages of the modified MK-40/PANI membrane for scaling mitigation?
Response
Thank you for this comment. The separation of monovalent and divalent ions is one of the possible applications of such modified membranes as MK-40/PANI. It is well known that multicharged ions form sparingly soluble and/or insoluble precipitates on the surface of the ion-exchange membrane at currents higher the limiting current as a result of water splitting reaction at the membrane/solution interface. Therefore, the identification of the effect of polyaniline on the process of water splitting is an important issue of fundamental and applied importance, which was the purpose of this study. Yes, unfortunately, it turned out that the precipitation on the surface of the membrane coated with a layer of polyaniline is much stronger than on the pristine membrane. However, this fact, in our opinion, is not a complete obstacle to the use of such materials in electrodialysis, but rather imposes some restrictions on the current operating modes.
Minor concerns,
- In the abstract, the authors said “A comprehensive study of the polyaniline influence on mineral scaling”, but the influence of polyaniline on scaling was not even investigated. The authors just compared two kinds of membranes, how the effect of the addition of polyaniline on membrane scaling was not investigated.
Response
In the work, the composition of the model solution was chosen to ensure membrane scaling formation while developing coupled effects of concentration polarization. Comparative experiments were conducted for the pristine and modified membranes under the same conditions (solution composition, hydrodynamic regime, identical auxiliary membranes). Chronopotentiograms of both membranes were measured with simultaneous monitoring of the solution pH in the desalination chamber. Three regimes were selected. The first is when the precipitate should not form (underlimiting current mode). The second is when the current is equal to the limiting. And the last is when the coupled effects of concentration polarization develop, the main of which are electroconvection and water splitting. That is, when the conditions for precipitate formation are created. Thus, the authors consider that the effect of polyaniline located on the surface of the ion-exchange membrane on the process of precipitation under the conditions of overlimiting current regime has been studied, since the presence or absence of a modifier is the only one differing factor that could affect this process.
- In line 29-30, methods of electromembrane technology are the most effective, environmentally cleanest and least costly. The sentence is overstated. Electromembrane is one of effective but should not be the most effective. Please modify such kind of statement.
Response
In line 30-31 we have made the corrections:
“To date, methods of electromembrane technology are effective, environmentally cleanest and least costly [1,2].”
- The composition of feed solution directly the membrane scaling. Why this kind of composition is chosen? According to the solubility product of CaCO3, it seemed that the feed solution is supersaturated. Please clarity it.
Response
The model salt solution was composed of Na2CO3 (1000 mg/L), KCl (800 mg/L), CaCl2*2H2O (4116 mg/L), and MgCl2*6H2O (2440 mg/L) to respect a Mg/Ca ratio of 2/5 to ensure membrane scaling formation. Several studies have been carried out in model salt solutions containing approaches to the bovine milk mineral composition [Casademont, C.; Farias, M.; Pourcelly, G.; Bazinet, L. Impact of electrodialytic parameters on cation migration kinetics and fouling nature of ion-exchange membranes during treatment of solutions with different magnesium/calcium ratios. J. Memb. Sci. 2008, 325, 570–579, doi:10.1016/j.memsci.2008.08.023], such as mixtures of calcium chloride and sodium carbonate and also the addition of magnesium chloride [Casademont, C.; Pourcelly, G.; Bazinet, L. Effect of magnesium/calcium ratio in solutions subjected to electrodialysis: Characterization of cation-exchange membrane fouling. J. Colloid Interface Sci. 2007, 315, 544–554, doi:10.1016/j.jcis.2007.06.056; Casademont, C.; Sistat, P.; Ruiz, B.; Pourcelly, G.; Bazinet, L. Electrodialysis of model salt solution containing whey proteins: Enhancement by pulsed electric field and modified cell configuration. J. Memb. Sci. 2009, 328, 238–245, doi:10.1016/j.memsci.2008.12.013]. These studies have continuously reported an undesirable mineral fouling formation on CEMs and AEMs, mainly when using solutions containing a high Mg/Ca ratio such as in milk, hindering the good performance of ED processes, especially when using a basified concentrate solution [Casademont, C.; Pourcelly, G.; Bazinet, L. Effect of magnesium/calcium ratio in solutions subjected to electrodialysis: Characterization of cation-exchange membrane fouling. J. Colloid Interface Sci. 2007, 315, 544–554, doi:10.1016/j.jcis.2007.06.056]. To prevent precipitation of calcium carbonate, the pH of the initial solution was adjusted to 6.5 by adding HCl. As is known, carbonate ions are absent in the solution at such pH value. Therefore, the precipitate does not form in the initial soution.
- For the current density, the authors sometimes used an exactly value such as i=5.0mAcm-2, but sometimes used a relative value of limiting current density i=1.4 ilim. Please make consistence of the description of the current density.
Response
Thank you for this suggestion. You are right. We have made consistence of the description of the current density. We have changed figure 6 and corrected the text in lines 284-285 and 344-345.
Figure 6. Chronopotentiograms of МK-40 and МK-40/PANI membranes in the model solution and corresponding time dependence of the pH of desalinated solution at i = 0.7ilim (a); i = ilim (b); i = 1.2ilim (c); i = 1.4ilim (d).
- In Line 298-300, “According to the energy dispersive analysis the precipitate on the membrane surfaces is a mixture of calcium carbonate and magnesium with a predominance of calcium carbonate”. How to make sure the precipitate is a mixture of calcium carbonate and magnesium with a predominance of calcium carbonate? Please provide evidences.
Response
Thank you for this question. The authors agree that it is an important part of the study. We add some evidences in the supplementary materials and correct the text in line 319-322:
“According to the energy dispersive analysis the precipitate on the membrane surfaces is a mixture of calcium carbonate and magnesium hydroxide with a predominance of calcium carbonate (Figure S1). There were similar findings obtained by C. Casademont et al. [52–55].”
